# T-Cell Response to Viral Hemorrhagic Fevers

**DOI:** 10.3390/vaccines7010011

**Published:** 2019-01-22

**Authors:** Federico Perdomo-Celis, Maria S. Salvato, Sandra Medina-Moreno, Juan C. Zapata

**Affiliations:** 1Grupo Inmunovirología, Facultad de Medicina, Universidad de Antioquia, Medellín 050010, Colombia; fcelis@ihv.umaryland.edu or federico.perdomo@usco.edu.co; 2Institute of Human Virology, School of Medicine, University of Maryland; Baltimore, MD 21201, USA; msalvato@ihv.umaryland.edu (M.S.S.); smmoreno@ihv.umaryland.edu (S.M.-M.)

**Keywords:** viral hemorrhagic fever, lassa virus, ebola virus, hantavirus, yellow fever virus, dengue virus, T-cells, vaccine, interferon-gamma, tumor necrosis factor-alpha

## Abstract

Viral hemorrhagic fevers (VHF) are a group of clinically similar diseases that can be caused by enveloped RNA viruses primarily from the families *Arenaviridae*, *Filoviridae*, *Hantaviridae*, and *Flaviviridae*. Clinically, this group of diseases has in common fever, fatigue, dizziness, muscle aches, and other associated symptoms that can progress to vascular leakage, bleeding and multi-organ failure. Most of these viruses are zoonotic causing asymptomatic infections in the primary host, but in human beings, the infection can be lethal. Clinical and experimental evidence suggest that the T-cell response is needed for protection against VHF, but can also cause damage to the host, and play an important role in disease pathogenesis. Here, we present a review of the T-cell immune responses to VHF and insights into the possible ways to improve counter-measures for these viral agents.

## 1. Introduction

Viral hemorrhagic fevers (VHF) constitute a group of clinically similar diseases characterized by mild to severe febrile acute syndromes with vascular damage, plasma leakage and bleeding. The most clinically prevalent human hemorrhagic fever viruses belong to the families *Arenaviridae*, *Filoviridae*, *Hantaviridae*, and *Flaviviridae* [1,2], which share some structural and replicative characteristics (Figure 1), as well as some immunity issues in the host. Indeed, the families *Arenaviridae* and *Hantaviridae* now belong to the same order: *Bunyavirales* [2,3]. However, the pathogenesis and role of the immune system in the development of mild or severe disease, and the protective or pathogenic function of some components of the immune system are still not clear. Particularly, the contribution of T-cell mediated immunity in disease protection and pathogenesis is yet to be characterized, especially the important associations between the magnitude of the T-cell response with disease survival or disease exacerbation. Thus, apparently, T-cells play a context-dependent role during VHF that is necessary to define in order to improve the patient clinical outcome and to develop vaccines or immunotherapeutic strategies. Here, we discuss the role of T-cells in the control of VHF and disease pathogenesis and their potential contribution to vaccine development. T-cell response features of representative viruses from each family, such as lassa virus (LASV) for arenavirus, ebola virus (EBOV) for filovirus, Hantaan Virus (HTNV) for hantavirus, and dengue virus (DENV) for flavivirus, are addressed. Since studies on vaccination against yellow fever virus (YFV) have largely contributed to the understanding of the dynamics of human T-cell differentiation after viral infections, and a large body of evidence supports the role of T-cells in the effectiveness of this vaccine, here we also discuss the T-cell response after YFV vaccination.

## 2. Epidemiology 

VHFs can be transmitted person-to-person through direct contact with contaminated body fluids or tissues. Most of the VHF are also zoonotic in nature, spreading through different mechanisms depending on each virus. Some of them are transmitted through the consumption of raw meat or fluids from infected animals, direct contact with rodents or bats, inhalation or contact with materials contaminated with rodent excreta. Others are vector-borne diseases, transmitted by bites of infected mosquitoes or ticks [4]. These diseases are clinically dynamic, ranging from asymptomatic to severe disease and death, rapidly progressing through several stages in a few hours or days. The absence of highly specialized laboratories and trained personnel limit the capacity for an early diagnosis of VHF, particularly in endemic areas [5]. Similarly, the lack of approved specific therapy contributes to disease burden and unfavorable clinical outcomes [6]. 

The epidemiology of VHF is highly variable, considering the geographic distribution of natural reservoirs or vectors around the world, and different syndromes caused by each of these viruses [7]. Lassa fever is endemic in West Africa, where 100,000–300,000 clinical infections are estimated each year, with 1–5% mortality [8]. However, some nosocomial outbreaks have occurred with a fatality rate of 50% [9]. In addition, case fatality rates reach 69% among patients who assisted at health care facilities (representing the most severe LASV infections) [10]. EBOV causes sporadic outbreaks in Central and West Africa, with a reported fatality rate of more than 40% in the majority of outbreaks [11]. A recent outbreak occurred in the Democratic Republic of the Congo. As of 27 December 2018, 543 cases had been confirmed, with a fatality rate of 57% [12,13]. Hantaviruses of the Old World such as HTNV and Puumala virus, cause hemorrhagic fever with renal syndrome (HFRS) mainly in China, Korea, and North-Eastern Europe. Hantaviruses of the New World such as Andes and Choclo virus, cause hantavirus pulmonary syndrome (HPS), with cases reported in the United States, Canada, Brazil, Chile, Argentina, and Panama. There are roughly 30 cases per year of HPS in the United States, with 35% case fatality [14]. The severity and mortality associated with these Hantavirus syndromes vary according to the causative virus, and death from HFRS is usually due to renal insufficiency, shock, or severe hemorrhage [15].

Yellow fever is endemic in tropical areas of sub-Saharan Africa and Central and South America. Despite having an effective vaccine, the coverage is low, particularly in some areas of Africa. The introduction of unvaccinated individuals to areas harboring mosquitoes of the genus *Aedes* leads to dissemination of this virus and major outbreaks [16]. Indeed, in endemic areas of Africa, approximately 1% of severe hepatitis may be caused by YFV infection [17]. For the year 2013, 130,000 cases of yellow fever were estimated, including 78,000 deaths [18], and in 2016, nearly 1000 cases were confirmed in Africa [19]. Moreover, an outbreak occurred in Nigeria in 2017, with 33 confirmed cases and a fatality rate of 27.3% [20]. Fewer cases are estimated to occur in America due to a higher vaccine coverage. However, yellow fever outbreaks occur in several countries, such as in Brazil, where, between 2017 and 2018, 464 yellow fever cases and 154 deaths were confirmed [21]. Individuals traveling to endemic areas are also highly affected by Yellow fever [22]. Dengue fever is a major public health concern, with at least 3.97 billion people at risk of infection [23]. Its vector, mosquitoes of the genus *Aedes*, is distributed in tropical and sub-tropical urban and rural areas [24]. Any of the four DENV serotypes can cause dengue fever, which is clinically classified as dengue and severe dengue. Dengue in turn can be subdivided into dengue without warning signs or with warning signs [25]. At least 390 million DENV infections are estimated per year. Although less than 25% of them are clinically symptomatic [26], at least 500,000 severe dengue cases and 20,000 related deaths are estimated to occur annually [27]. Thus, VHF are important health concerns worldwide, and improvement of prevention, diagnosis, and treatment options are required to limit the impact of these diseases.

## 3. Pathogenesis and Common Characteristics of the T-Cell Response Against VHF

There is a strong correlation between viral loads and the severity of the disease, suggesting a failure of the initial host immune responses [28]. A major similarity between VHF entities is the context-dependent protective and/or detrimental role of T-cells (Table 1 and Figure 1). Indeed, VHF are characterized by a massive T-cell activation, differentiation to effector profiles, and predominance of type-I effector mechanisms (production of Interferon [IFN]-γ and cytotoxicity). Several markers have been used for the study of the T-cell response against natural and vaccine challenges with hemorrhagic fever viruses (Table 2). The dominant activation phenotype is the HLA-DR^+^ CD38^+^, which characterizes effector T-cells [29,30]. Due to the low preimmunization levels of this phenotype, it allows us to identify the expansion of activated T-cells with natural infection or vaccination [29]. Both CD4^+^ and CD8^+^ T-cells are important for protection against VHF, and their effective stimulation by immunodominant epitopes, as well as the activation of their lytic and non-lytic effector mechanisms, are required for natural or vaccine-mediated protection. CD4^+^ T-cells are also important for the control of some VHF through the stimulation of B-cells for neutralizing antibody production [31], and promoting the CD8^+^ T-cell response during chronic infections [32]. On the other hand, a pathogenic role for T-cells has been demonstrated for some VHF, in which an exacerbated T-cell response contributes to organ damage and fatal disease, as well as sub-acute/chronic inflammatory clinical conditions [33,34,35]. Particularly, T-cells have been purported to contribute to the development of two hallmarks of VHF: endothelial damage [35,36], leading to vascular leakage, and hepatic compromise, which contributes to coagulation disorders [37,38,39,40,41,42]. Of note, several studies have shown associations between the magnitude, phenotype, and/or function of T-cells and disease outcome in several VHF (survivors and non-survivors). Non-survivor patients may not generate an efficient immune response in comparison with survivors, in whom the response is early and robust. In addition, defects in the regulation of T-cell homeostasis are associated with severe disease and fatalities [43,44]. However, it has to be considered that disease non-survivor characteristics are less known due to the limitation of their inclusion in these studies, as disease complications and death are rapidly developed [4].

Due to the multi-organic damage and hemodynamic compromise, VHF differ from diseases caused by other ssRNA viruses. Indeed, clinically relevant ssRNA viruses include respiratory syncytial virus, influenza virus, human metapneumovirus, rhinovirus, coronavirus, and parainfluenza virus, which are transmitted via the respiratory mucosa and cause a variety of pulmonary syndromes [59]. In these diseases, lung-confined dendritic cells present viral antigens to T-cells located in regional lymph nodes. Effector T-cells are then recruited into airways, lung interstitium and vasculature, where they are required for efficient viral clearance [60]. In addition, antigen-specific T-cells can be found in the lungs following recovery from respiratory ssRNA virus infection [61], suggesting that a compartmentalized T-cell response plays a key role in protective immunity to respiratory virus infections. Nonetheless, massive T-cell activation can be observed in influenza A virus infection with systemic compromise [62]. Thus, the T-cell response can vary between clinically relevant hemorrhagic fever and respiratory ssRNA viruses, differing primarily in the systemic/multi-organic compromise and effects on vasculature. For instance, filoviruses cause the infected cells to secrete TNF-α which directly increases gaps between endothelial cells causing vascular leakage [63] but Lassa suppresses TNF-α production in the infected cell but indirectly stimulates it in uninfected bystanders [64]. The HF-causing LCMV-WE also elicits high IL-6 and TNF-α primarily in uninfected bystander cells, that eventually contributes to vascular leakage [39,65,66].

Throughout the course of VHF, a particular pattern and kinetics of circulating cytokines is apparent (Figure 2). Early during the acute phase of infection (between the first and fourth day after the onset of symptoms), coinciding with the peak of viremia and development of fever, there is an increase of IFN-α and β, inflammatory cytokines such as tumor necrosis factor (TNF)-α and interleukin (IL)-6 [33,67,68,69,70], oxygen and nitrogen reactive species [71], lipid mediators and other soluble factors [72]. These and other mediators of inflammation are produced by antigen-presenting cells, endothelial cells, polymorphonuclear cells, NK cells, hepatocytes, among others [73], and are known to promote the polarization of T-cells, which together with the antigen presentation and costimulatory molecules, induce the activation and differentiation of T-cells to effector profiles [74]. Consequently, the levels of T-cell-derived IFN-γ and IL-2 increase between the 5th and 10th day after the onset of symptoms, a period displaying the most of severe disease manifestations [67,75]. The regulatory cytokine IL-10 also reaches a peak during this phase [67,69,70]. Together, the massive release of inflammatory and regulatory cytokines is known as a “cytokine storm”, which is a hallmark of VHF and is associated with severe disease [36,76,77,78]. Generally, 12 to 14 days after the onset of symptoms, the cytokines return to basal levels, coinciding with the recovery or convalescence phase. In some cases, the persistence or reactivation of IFN-γ- and/or TNF-α-producing activated T-cells is observed after 40 days of the onset of symptoms, which may explain the development of post-VHF syndromes [33,34,35].

### Regulatory and Other T-Cell Subsets in VHF 

There is a large gap in our understanding of the role of T-cell subpopulations in the pathogenesis of VHF. Among them, regulatory T-cells are an important subset, as they are essential for the control of exacerbated immune responses and limit chronic inflammation [79]. Strikingly, the frequencies of circulating regulatory CD4^+^ T-cells remain unchanged in EBOV, hantaviruses and severe DENV infections [28,80,81,82,83], which may account for the massive T-cell activation, chronic inflammation, and disease severity observed in these infections. Unchanged or decreased proportions of regulatory CD4^+^ T-cells can be observed after immunization with the YF-17D vaccine, but this subset is activated after vaccination and its frequency positively correlates with the magnitude of total activated T-cells [84,85], consistent with the induction of regulatory responses in the setting of high T-cell activation. Other T-cell regulatory mechanisms, such as the expression of the inhibitory receptors cytotoxic T-lymphocyte antigen 4 (CTLA-4) and programmed death 1 protein (PD-1) [43,46,80,86], as well as high levels of regulatory cytokines [70,87], would exert a regulatory role in these diseases (discussed below). Other T-cell subsets, such as γδ T-cells, have also been evaluated in some VHF, with the proposed anti-viral role of bridging innate and adaptive immunity, particularly through the production of IFN-γ [88]. Finally, specific effector T-cell profiles, such as IL-17-producing cells and related regulatory genes, are increased during challenges with hemorrhagic fever viruses, and have been implicated in protective functions [31,76,89,90,91,92]. Nonetheless, more studies are required to define the role of these non-conventional T-cell subsets in VHF.

## 4. Particular Characteristics of the T-Cell Response Against VHF

### 4.1. T-Cell Response Against LASV

A large body of evidence supports a critical role for T-cells in the control of LASV infection, especially if the poor humoral response elicited during this infection is considered [93,94]. However, LASV induces profound immune defects, both at the cellular and organ level. Early reports showed that nonhuman primates with fatal LASV infection exhibited a poor T-cell response to mitogens [95]. Viral replication and damage in secondary lymphoid organs and pronounced lymphopenia are also observed in fatal LASV infection [52,96,97]. Moreover, early low and delayed activation of T-cells, low cytokine production, and high viral loads are associated with fatal LASV disease [97,98], consistent with inadequate activation signals provided by antigen-presenting cells. Certainly, LASV infects human monocyte-derived dendritic cells and impairs their cytokine secretion, expression of costimulatory molecules and T-cell-stimulation capacity. Under these conditions, T-cells are unable to differentiate into effector or memory cells that control viral replication [56]. HLA class II and key T-cell transcription factor genes are also downregulated in cells from viremic lymphocytic choriomeningitis virus (LCMV)-WE-infected macaques [65], a model for pathogenic LASV infection of humans [99]. These dendritic cell defects appear to be disease-specific, as infection of human dendritic cells with the closely-related nonpathogenic Mopeia virus (MOPV) elicits an early and strong T-cell response [64,100]. In addition, a cross-talk between dendritic cells and T-cells may be required for efficient cellular immune responses [100]. Thus, the impairment of early T-cell activation may account for the unchecked viral replication and severity of LASV disease, in contrast to nonpathogenic arenaviruses. Other alterations in T-cells observed in the LCMV-WE macaque model include the decrease in γδ T-cells possibly through cytokine-mediated apoptosis [101]. Finally, some differences can be found between the immune response against Old and New World arenaviruses (reviewed in [66]). As mentioned, LASV infection is characterized by immune suppression, whereas the New World arenavirus Junin infection proceeds with a massive cytokine secretion [102]. In addition, apparently, the T-cell may be less critical than antibody response in protection against Junin virus infection [103].

Despite the aforementioned impairment in the T-cell response during LASV infection, evidence of their protective role in this disease include: (i) presence of a strong and partially cross-reactive T-cell response in individuals who live in areas of LASV endemicity, which may exert protection against viral re-exposure [47]; (ii) association between the magnitude and function of LASV-specific T-cells and viral clearance in a non-fatal human infection [33]; (iii) survival of LASV-infected macaques associated with the activation, proliferation and cytokine production by T-cells in response to LASV antigens [97]; (iv) T-cell-mediated protection by LASV vaccine candidates in nonhuman primates [98,104,105,106] and rodents [107,108]; (v) T-cell-mediated protection against lethal arenavirus challenge in nonhuman primates previously exposed to pathogenic arenaviruses [109]. During LASV infections in macaques, an increase in the proportion of activated CD69^+^/CD25^+^, and proliferating Ki-67^+^ T-cells is observed, particularly in nonfatal disease. A decrease in the proportion of CD28^+^ T-cells is also observed in LASV-infected macaques, consistent with differentiation of effector cells [97]. Similarly, in surviving human beings, LASV infection corresponds to a large increase in the proportion of activated HLA-DR^+^ CD38^+^ T-cells, with a Ki-67^+^ Bcl-2^−^ PD-1^+^ CD45RA^−^ phenotype, consistent with an effector memory profile. LASV-specific CD8^+^ T-cells also exhibit an effector memory phenotype. Interestingly, the peak of LASV-specific CD8^+^ T-cells, as well as polyfunctional cytokine CD4^+^ T-cell responses, are observed 20 days after symptoms onset, coinciding with viral clearance [33].

Remarkably, because major histocompatibility complex (MHC) class I knock-out mice exhibit persistent LASV replication (consistent with an important role of CD8^+^ T-cells in LASV control), T-cells continue to encounter LASV epitopes and stimulate monocyte/macrophages to secrete proinflammatory cytokines, thereby contributing to prolonged pathogenesis [77]. Certainly, in the convalescence phase of human LASV infection, antigen-specific CD8^+^ T-cells exhibited a high production of IFN-γ and TNF-α, which associated with the development of lymphadenitis, epididymitis, and chills, all in the absence of detectable viremia [33]. These findings support the hypothesis of LASV antigen persistence, the consequent activation of CD8^+^ T-cells, and a possible immune-mediated pathology.

Some T-cell epitopes have been described for LASV proteins, particularly in the nucleoprotein. Ex vivo stimulation with LASV nucleoprotein peptides elicits a strong proliferative and cytokine response in human T-cells from individuals living in LASV endemic areas [47]. LASV nucleoprotein peptides predicted by computational analysis also induce CD8^+^ T-cell responses in HLA-A2.1 transgenic mice [110]. In addition, up to 1% of CD8^+^ T-cells from an acute LASV-infected patient were found to be specific for glycoprotein tetramers [33]. CD8^+^ T-cells from HLA-A2.1 transgenic mice recognize peptides of the LASV glycoprotein precursor and exert protection against LASV challenge [111], and CBA/J mice immunized with the MOPV-LASV reassortant virus also exhibit an IFN-γ response against the LASV glycoprotein [107]. Similarly, human CD4^+^ T-cells recognize a highly conserved 13 amino acid peptide located in the N-terminal part of LASV glycoprotein 2, with strong proliferation and IFN-γ production after ex vivo stimulation [112]. Vesicular stomatitis virus vectors expressing the LASV glycoprotein or modified vaccinia Ankara-LASV producing virus-like particles also elicit T-cell IFN-γ production to this protein and protect nonhuman primates and mice from LASV challenge [98,113]. Thus, vaccine candidates should be able to induce T-cells specific for LASV nucleoprotein and glycoprotein for efficient long-lasting protection. 

### 4.2. T-Cell Response Against EBOV Infection

Comparisons of EBOV survivors and fatalities have revealed that disease survival is associated with an early detection of EBOV-specific T-cells [43] and sustained T-cell cytokine expression [114], whereas fatal EBOV cases are associated with high expression of the inhibitory receptors CTLA-4 and PD-1, as well as with increased lymphocyte apoptosis [115]. Certainly, EBOV disease is characterized by apoptosis of T-cells [116], which has been attributed to activation-induced cell death mechanisms, direct lysis via EBOV GP [117], and T-cell dysfunction [53]. EBOV also affects T-cell activation by human dendritic cells [56]. These mechanisms prevent the development of an efficient adaptive T-cell response in most EBOV-infected individuals [118]. 

Importantly, in murine studies, CD8^+^ T-cells appear to play a major role in the control of EBOV infection. Evidence supporting this idea includes: (i) The adoptive transfer of CD8^+^ T-cells from mice infected with a mouse-adapted EBOV confers protection to EBOV-challenged naïve mice [118]; (ii) CD8^+^ T-cell-deficient, but not CD4^+^ T-cell- or B-cell-deficient mice, die after acute infection with a mouse-adapted EBOV [119]; (iii) CD8^+^ but not CD4^+^ T-cells are required for protection after EBOV-like particle mouse immunization. In accord with these findings, the cytokine response to EBOV peptide stimulation is predominantly found in human CD8^+^ T-cells [28,48].

In human beings who survive disease, EBOV infection induces a massive activation and terminal differentiation of CD8^+^ T-cells, with increased frequencies of activated HLA-DR^+^ CD38^+^ cells, proliferating Ki-67 cells, CD45RA^−^ CCR7^−^ (effector memory) and CD45RA^+^ CCR7^−^ (terminal effector memory) cells, even after 59 days of illness, when viral components are not found in blood by conventional methods [28,48]. On the other hand, excessive T-cell activation has been associated with fatal EBOV infection in humans [44]. Consistent with the expansion of effector memory cells, CD8^+^ T-cells from EBOV-infected patients are also equipped with a high content of granzyme B and have low expression of Bcl-2. In addition, these cells express the CX3CR1 receptor, suggesting that they could home to inflamed tissues [48]. Apparently, long-lasting activated T-cells are maintained by the persistent low-level antigens in sanctuary sites, particularly after moderate to severe disease, as has been found in eyes, brain and testes from EBOV-infected rhesus monkeys [120], and in urine and sweat from a severely ill patient [121]. Indeed, patients with mild disease and rapid viremia clearance, have a faster contraction of the activated T-cell response [48]. The persistent antigen load and T-cell chronic activation could contribute to the so-called post-EBOV disease syndrome, characterized by musculoskeletal pain, headache, and ocular problems [34]. 

Through ex vivo peptide stimulation to elicit cytokine production, the proteins targeted by human EBOV-specific T-cells have been characterized. The response to viral nucleoprotein (NP) (the most abundant and conserved protein among filoviruses [122]) is present in the majority of patients, with the induction of IFN-γ and TNF-α [48,50]. Reactions to peptides derived from Ebola glycoprotein (GP) and viral protein 24 (VP24), VP30, VP35, and VP40 are more variable among individuals, and of lower magnitude [28,48,50]. Importantly, most of the vaccine candidates against EBOV, such as cAd3-EBOV, rVSV/ZEBOV and Ad26.ZEBOV, only incorporate the EBOV GP in their vectors, possibly impairing the induction of a protective T-cell response [123]. Indeed, strong and protective CD8^+^ T-cell responses are induced after immunization with EBOV NP-expressing vectors or NP-DNA vaccines in mice [124,125,126,127,128]. Vaccination of nonhuman primates with EBOV-like particles induces strong NP-specific T-cell responses that are at least partially responsible for disease protection [129]. In addition, adoptive transfer of NP-specific CD8^+^ T-cells protects EBOV-challenged mice [124]. Together, these data highlight the importance of NP for an effective T-cell response to EBOV vaccines. Furthermore, we speculate that the success of the NP-containing vectors relies on the fact that NP is contributing structurally-stable particles to the vaccine, which are known to efficiently boost immunity [93]. A good example is the recent success of an EBOV vaccine that expresses the EBOV GP and VP40, that form virus-like particles in vivo and successfully protect NHP from lethal challenge [130].

Although CD4^+^ T-cells are apparently dispensable during an acute EBOV infection, they may play an important role in the stimulation of B-cell responses and anti-EBOV antibody production, which are important correlates of EBOV protection in vaccine studies [131,132]. For instance, the frequency of circulating CXCR5^+^ CD4^+^ T-cells (the blood counterparts of follicular CD4^+^ T-cells [133]) correlates with antibody titers induced by the rVSV-ZEBOV vaccine candidate in humans [31]. In addition, the follicular CXCR5^+^ CD4^+^ T-cell subset is expanded after filoviruses-like particle vaccination in mice [134,135], and this population supports B-cell antibody production via CD40L-CD40 interactions [135]. Thus, the stimulation of follicular CD4^+^ T-cells should also be targeted by EBOV vaccine candidates.

### 4.3. T-Cell Response Against Hantaviruses 

Although important gaps remain in our understanding of the T-cell response against Hantaviruses, most of the available data indicate that they are associated with disease protection. First reports indicated a role of T-cells in the resistance of mice to HTNV infection, as HTNV-challenged mice were protected after adoptive transfer of complete immune spleen cells, but not of T-cell-depleted cells [136,137]. HTNV-specific T-cells are induced in patients with HFRS, and their frequencies are higher in patients with a mild/moderate disease, compared with severe/critical HFRS patients [45,51,138]. Clinical remission after severe HFRS is also associated with a sustained IFN-γ/granzyme B CD8^+^ T-cell response, even in the absence of detectable neutralizing antibodies [139]. In contrast, a low IFN-γ T-cell response is also associated with severe HFRS [140]. Among total T-cells, the CD8^+^ subset is responsible for the major response to HTNV peptides [45,140]. Indeed, HTNV-specific CD8^+^ T-cells are required for efficient viral clearance in mice via production of IFN-γ, TNF-α, and cytotoxicity, and their low frequency leads to a persistent infection [141,142], such as seen in the hantavirus reservoir, deer mice [143]. In addition, hantavirus nucleocapsid protein inhibits the activity of granzyme B and caspase 3-mediated apoptosis of infected endothelial cells, consistent with an immune evasion mechanism of these viruses and explaining their persistence in rodent reservoirs [57]. CD4^+^ T-cells are also important for HTNV infection control and favorable clinical outcome, through polyfunctional cytokine responses and cytotoxic mechanisms [51]. Of note, in comparison with hantaviruses, Crimean-Congo hemorrhagic fever virus (CCHFV) and Rift valley fever virus (RVFV) are arthropod-borne diseases, where the route of transmission could influence the T-cell response. However, the available data indicate that T-cells undergo a similar activation profile in CCHFV-challenged humanized mice and RVFV-challenged C57BL/6 mice [144,145], T-cells are important for protection after RVFV challenge in mice [145,146], whereas in human beings, these infections can induce long-lived memory CD8^+^ T-cells [147]. Finally, it remains to be determined if there are differences in the T-cell response against Old and New World hantaviruses, that may account for the different clinical syndromes found after these infections.

Similar to other VHF, a massive increase in the frequency of circulating Ki-67^+^ HLA-DR^+^ CD38^+^ CD8^+^ T-cells is observed in acute human hantaviruses infections, followed by a contraction phase which coincides with viral clearance. These cells exhibit low expression of CCR7, CD45RA, and CD127, and high expression of perforin and granzyme B, consistent with an effector memory profile [80]. Inverse CD4:CD8 ratios are also observed during HFRS [148]. Interestingly, human CD4^+^ and CD8^+^ T-cells display a differential pattern of expression of CTLA-4 and PD-1 during acute hantaviruses infections. Thus, CD4^+^ T-cells have high expression of both inhibitory receptors, whereas CD8^+^ T-cells only express CTLA-4 [80], which may account for the more efficient CD8^+^ T-cell responses in these infections, but also a tendency for immunopathology (discussed below).

Several reports have evaluated the human T-cell epitopes on hantaviruses. In patients with HPS caused by Sin Nombre Virus (SNV), CD8^+^ T-cells recognize epitopes on the viral nucleocapsid protein restricted by HLA-C7 and HLA-B35, whereas CD4^+^ T-cells recognize epitopes on the viral nucleocapsid protein restricted by HLA-DQw2 or DR3 [149]. HTNV nucleocapsid protein epitopes restricted by HLA-A1, A2, and B51 are also recognized by T-cells in patients with HFRS [45,138], or in individuals with laboratory-acquired HTNV infections [150]. The recognition of nucleocapsid protein is expected, as several hantaviruses share conserved sequences of this protein. However, clinically relevant viruses, such as Puumala virus, exhibit nucleocapsid protein sequence variants and different immunodominant epitopes, that may evade efficient T-cell responses or limit cross-protection among hantaviruses [149,151]. In addition, human T-cells also recognize epitopes on hantaviruses glycoproteins [35,139,150,152,153]. Indeed, the glycoprotein contains an amino acid tail that selectively directs its proteasomal degradation, which would increase its presentation by the HLA class I pathway [154].

Consistently, the immunization of Syrian hamsters with adenoviral vectors expressing the Andes virus nucleocapsid or glycoprotein, protected them against a viral challenge, with an important contribution from cytotoxic T-cells [155]. A similar protection was observed in bank voles immunized with Puumala virus nucleocapsid protein [156,157]. Strikingly, immunization of bank voles with nucleocapsid protein from different hantavirus strains elicits cross-protective T-cell responses against Puumala virus [158]. Collectively, these studies underline the importance of T-cell immunity for protection against hantavirus infections and support the relevance of nucleocapsid protein for the generation of cross-protection using novel vaccine candidates.

Importantly, although T-cells have been associated with hantaviruses disease protection, a role in immunopathology has also been suggested. HTNV-induced meningoencephalitis in mice has been associated with T-cell mediated immunity [159]. Some HLA class I alleles, particularly the B8, have also been associated with severe Puumala virus infection [160,161]. Similarly, patients with fulminant HPS exhibit higher frequencies of SNV-specific CD8^+^ T-cells compared with patients with moderate disease, and their excessive production of TNF-α and IFN-γ likely induces damage of pulmonary endothelial cells, contributing to the observed capillary leakage during HPS [35]. Indeed, fatal HPS is characterized by an increased frequency of cytokine-producing cells in the lungs [162]. The unchanged frequencies of regulatory T-cells, lower expression of inhibitory receptors in CD8^+^ T-cells [80], and decreased levels of the regulatory cytokine IL-10 [163], could also contribute to an exacerbated T-cell response during hantavirus infection. 

In summary, the magnitude and quality of the T-cell response, as well as host characteristics, determine the final outcome of hantavirus infections. Studies that define these immune features are required to improve disease management and prognosis, as well as to contribute to vaccine development.

### 4.4. T-Cell Response after YFV Vaccination

Yellow fever is a prototypic disease for the study of the T-cell responses elicited after viral infections and vaccination (Figure 3). The YF-17D vaccine is a live attenuated vaccine, containing both structural and non-structural (NS) proteins of YFV, and is one of the most safe and effective vaccines currently available, inducing seroconversion and memory T-cell responses in more than 90% of individuals in most of the studies, with neutralizing antibodies at protective levels and long-lived memory T-cells for more than 10 years [164,165]. YF-17D interacts with, and replicates, in dendritic cells, stimulating the expression of MHC and costimulatory molecules, the production of inflammatory cytokines and the presentation of dominant epitopes, which together promote efficient T-cell priming [166,167,168,169]. Thus, the T-cell response generated after YF-17D immunization complements the neutralizing antibody response, which is critical for YF-17D-induced protection [164]. Evidence of the protective effect of the T-cell response against YFV includes: (i) the depletion of CD8^+^ T-cells in YF-17D-vaccinated B-cell deficient mice abolishes the vaccine-induced control of an intracerebral viral challenge [170]; (ii) a polyfunctional CD4^+^ T-cell response after YF-17D immunization complements neutralizing antibodies for protection against a virulent YFV challenge in mice [171]; (iii) IFN-γ restricts YF-17D viscerotropic and neurotropic dissemination, and is a mechanism of in vivo attenuation of this vaccine [172]; (iv) cytokine-producing human CD4^+^ T-cells expressing CD40L, a molecule important for stimulation of B-cell responses [173], are associated with the levels of neutralizing antibodies after YF-17D immunization [168], and with vaccine-induced protection in mice [170].

After YF-17D vaccination, there is an increase in the levels of soluble markers of immune activation, such as β2-microglobulin and neopterin [174]. Accordingly, an increase in the frequency of human HLA-DR^+^ CD38^+^ T-cells is observed seven days post-vaccination (coinciding with virus clearance), with a peak at two weeks, and returning to pre-vaccination levels after one month. [29,84,175] (Figure 3). The peak of the CD4^+^ T-cell response appears to be earlier than that of CD8^+^ T-cells [84]. YFV-specific CD8^+^ T-cells follow a similar kinetics, but remain detectable 6–12 months post-vaccination [29,30]. In the acute phase, these cells exhibit an effector profile, with high expression of Ki-67, the chemokine receptor CCR5 (that allows the migration to inflamed tissues), granzyme B, and PD-1, and low levels of Bcl-2, CCR7 and CD45RA. This effector phase is followed by the acquisition of a memory profile, with inversion in the expression of the mentioned molecules [30,176]. A high expression of T-bet (transcription factor which characterizes effector cells) is also observed early after vaccination, with a shift to a predominant Eomesodermin expression (transcription factor which characterizes memory cells) three months post-immunization [176]. Interestingly, a low to undetectable bystander activation of CD8^+^ T-cells is observed after YF-17D vaccination [29], indicating that most of the increased effector cells are vaccine-specific. Remarkably, YF-17D induces a broad human T-cell response, with detectable cytokine production after stimulation with epitopes derived from each of the 10 YF-17D proteins [30,177]. This finding is consistent with the diverse T-cell receptor repertoire observed in YFV-specific T-cells post-vaccination [178,179]. Nonetheless, the major antigen-specific population is directed against epitopes from the E, NS1, NS2B, NS3, and NS5 proteins [30,84,180]. The magnitude of the T-cell response is also associated with the “promiscuous” binding of YF-17D epitopes to HLA class II molecules [181]. Remarkably, the T-cell response elicited by YF-17D is polyfunctional [30,84,168,177], whereas different effector T-cell profiles, such as type-I and II effector mechanisms can be observed [167,182,183,184], depending on the pathogen-recognition receptors activated on antigen-presenting cells [167].

Along with the robust effector T-cell response early after vaccination, long-lived memory T-cells are also induced by YF-17D (Figure 3). YFV tetramer^+^ cells are detectable 18–25 years after a single vaccination [165,176], with robust memory T-cell responses being recalled after ex vivo stimulation [30,165,175]. A recent study described the turnover of human memory YFV-specific CD8^+^ T-cells induced after YF-17D vaccination [185]. Vaccinated individuals received deuterated water at different time points to evaluate its incorporation into DNA as a measure of cell replication. A low proliferation of memory CD8^+^ T cells was observed after the effector phase (where a sequential expansion and contraction in the number of effector cells occurs), dividing only once each 485 days, but with survival for more than 2 years after immunization. These long-lived memory CD8^+^ T cells retain an “epigenetic memory” of effector cells, and are poised to exert effector functions with re-stimulation [185]. Interestingly, neither the magnitude nor the quality of the YFV-specific CD8^+^ T-cell response is influenced by YF-17D vaccine boosters [175,176] (Figure 3). This finding is explained by the antibody neutralization and rapid clearance of viral load during secondary immunization, that prevents the efficient stimulation of the immune system [175]. Indeed, the magnitude of the human CD8^+^ T-cell response after primary YF-17D immunization is highly determined by the initial viral load [186]. These findings indicate that the T-cell-mediated immunity induced by live attenuated vaccines is dependent on the antigen levels reached after immunization, and this mechanism explains the highly protective immune responses elicited after a single immunization with the YF-17D.

### 4.5. DENV and Other Flaviviruses Elicit Cross-Reactive T-Cell Responses

Dengue fever is a typical disease where immune mechanisms determine the pathogenesis and clinical severity. The presence of four antigenically related serotypes, in addition to other closely-related flaviviruses (e.g., ZIKV) circulating in most of the endemic areas, and the cross-reactive but non-neutralizing immune responses against them, explain the higher frequency of severe disease after secondary infection with different DENV serotypes [187]. In addition, most of the severe manifestations occur following viral clearance, implying that the immune response elicited by the virus is a major cause of pathology [67,188]. In the case of the T-cell response against DENV, the excessive production of pro-inflammatory cytokines, known as the “cytokine storm”, is responsible for endothelial damage, vascular leakage and shock [36]. Accordingly, some HLA-A and B alleles [189,190,191,192,193], the timing of the T-cell response [67], the magnitude of their activation [46,194], and some of their effector molecules [195,196,197,198], have been associated with the development of severe dengue. Indeed, human T-cells are a major target of a multiplicity of stimulating factors and chemoattractants derived from monocytes, dendritic cells, and B-cells, inducing their massive activation during DENV infection [195,199,200,201,202,203]. Thus, serotype cross-reactivity of DENV-specific T-cells generated during primary infection, their exacerbated cytokine and lytic response inducing organ damage, competition with T-cells avid for the current infecting serotype, and enhancement of DENV infection of target cells, are mechanisms of the pathological role of T-cells during secondary DENV infections [204,205,206]. Nonetheless, the host factors that determine the dengue fever pathology and clinical outcome remain to be determined. Similar to other VHF, DENV infects antigen-presenting cells and affects their T-cell priming capacity [58,207,208,209]. This could be related to the delayed appearance of DENV-specific T-cells, which in turn is associated with the development of severe manifestations [210,211]. DENV infection also induces T-cell apoptosis [55] and affects their ex vivo response to polyclonal stimuli, which could confer susceptibility to co-infections [212].

A protective role for T-cell responses has also been demonstrated in DENV and other flavivirus infections, both after natural and vaccine challenges. A strong T-cell response is induced after DENV natural infection, exhibiting an effector memory phenotype, IFN-γ and TNF-α production, and cytotoxic potential [46]. In addition, DENV-specific T-cells home to the skin, a major site of DENV replication, through the expression of the cutaneous lymphocyte-associated antigen [213], suggesting an immune surveillance function of memory T-cells. In vivo, serotype-specific and cross-reactive CD8^+^ T-cells contribute to viral clearance after a DENV challenge in IFN-α/βR^−/−^ HLA-B0707 transgenic mice [214]. A high frequency of memory T-cells specific for conserved epitopes of DENV serotypes is associated with protection against secondary clinical infection in endemic areas [215,216], whereas individuals expressing particular protective HLA class I alleles exhibit strong CD8^+^ T-cell responses and protection against re-infection [217]. Moreover, individuals with an early DENV-specific T-cell IFN-γ response exhibit lower viremia and develop milder clinical disease [218]. Cytotoxic CX3CR1^+^ CD4^+^ T-cells also play a protective role during DENV infection [219] and immunization with CD4^+^ T-cell epitopes prior to infection contributes to viral clearance after subsequent viral challenge in IFN-α/βR^−/−^ mice [220]. HLA-DR allele associations with disease resistance [221,222], as well as a broad epitope reactivity of human CD4^+^ T-cells [223], support a protective role of CD4^+^ T-cells during DENV infections. Thus, there is evidence of T-cell-mediated protection against DENV infection, and T-cells have been proposed as immune correlates of protection in this disease [224].

Importantly, the antigenic relation of several flaviviruses allows T-cell-mediated cross-protection that is critical for a broadly protective flavivirus vaccine. Indeed, DENV-specific CD8^+^ T-cells, but not DENV-immune serum, confer cross-protection against subsequent Zika virus (ZIKV) infection in mice [225,226]. Immunization with cross-reactive ZIKV/DENV epitopes protects against a subsequent ZIKV challenge by a CD8^+^ T-cell-dependent mechanism in mice [227]. Sequential flavivirus immunizations also confer T-cell-mediated cross-protection against subsequent DENV infection in mice, whereas flavivirus cross-reactive human memory T-cells can be reactivated after heterologous virus exposure and exhibit an IFN-γ response [228]. Prior DENV exposure, both natural or after vaccination, also influences the timing and magnitude of the human T-cell response against subsequent ZIKV infection [229], pointing to a protective effect of viral exposure and improvement of the T-cell response.

There are at least 2,181 human T-cell epitopes identified in Flaviviruses, the majority of them in DENV [230]. Although human T-cell epitopes have been found in DENV structural proteins [46,49,231], the dominant responses are directed against non-structural proteins (NS), particularly NS3 [46,49,55,231,232]. Interestingly, DENV epitopes are differentially targeted between serotypes, with DENV-1, -2, and -4 eliciting an NS-specific CD8^+^ T-cell response, but DENV-3 inducing responses against the structural C, M and E proteins [233]. In addition, NS3 has important sequence homologies with ZIKV and other flaviviruses, highlighting the value of this protein in the induction of human T-cell responses [230].

Several vaccine candidates for DENV have been tested [234]. The most advanced is a tetravalent chimeric vaccine based on the live attenuated YF-17D vaccine (YF-17D-DENV vaccine), which incorporates the preM and E proteins of each DENV serotype, and the backbone and non-structural proteins of YFV. Similar to YF-17D, the tetravalent chimeric YF-17D-DENV vaccine has also demonstrated safety and immunogenicity in pre-clinical and clinical studies [235,236,237]. Remarkably, the efficacy results from phase 3 studies are not entirely explained by the magnitude of neutralizing antibodies, and cellular immunity could have a complementary role in this conferred protection [224]. Certainly, T-cell responses (particularly of CD8^+^ cells) are elicited by chimeric YF-17D-DENV vaccines. In YF-17D-DENV-2 immunized mice, T-cell responses against DENV-2 preM and E proteins are observed, conferring protection against subsequent DENV-2 intracerebral challenge, whereas T-cell depletion abrogates this protective effect [238]. However, human antigen-specific T-cells induced by the tetravalent YF-17D-DENV vaccine are directed against YFV, but not DENV NS3 protein [239]. Although it remains to be confirmed, this vaccine-induced T-cell response could exert a cross-protective role against DENV infection but could also promote suboptimal responses and immunopathology in secondary exposures (discussed below). This could explain the higher incidence of hospitalization for dengue fever in children younger than 9 years of age after vaccination with the tetravalent chimeric YF-17D-DENV vaccine [240]. Importantly, according to the immunodominance of NS3 protein, human vaccination with the TV003/TV005 vaccine, a live attenuated vaccine containing all DENV structural and non-structural proteins, elicits a strong polyfunctional NS3-specific response, comparable to that induced after natural infection [241,242]. These data support the inclusion of DENV NS proteins in vaccine candidates in order to improve the T-cell response, and possibly the protection conferred by this vaccine. 

Remarkably, an early and massive activation and differentiation of human effector CD8^+^ T-cells is observed during acute DENV infection, with HLA-DR^+^ CD38^+^ cells constituting more than 50% of the total circulating CD8^+^ T-cells in some individuals. However, only a small fraction of these activated CD8^+^ T-cells are DENV-specific [46], an observation that is related to the massive apoptosis of high-avidity antigen-specific T-cells during acute infection [55]. In addition, most of the DENV-specific T-cells are low-affinity serotype cross-reactive [231,243], and their expansion would impair the overall T-cell response and viral clearance [244]. Certainly, activation of high avidity antigen-specific T-cells may result in subsequent optimization of their signal-transduction machinery and a preferential IFN-γ production, compared with low avidity antigen-specific cells [245], which would confer a long-lasting protective function of T-cells during infection. Accordingly, children who maintain higher frequencies of IFN-γ- and IL-2-producing DENV-specific T-cells and get re-infected with DENV, experience a subclinical infection, compared with children who develop a symptomatic secondary infection and have lower frequencies of these antigen-specific subsets [216]. Genes related to antigen presentation, T-cell costimulation, T-cell receptor signaling, and IFN-γ-mediated signaling, are also over-expressed in patients with mild disease compared with those with severe dengue [246]. In contrast, serotype cross-reactive DENV-specific T-cells have lower degranulation but higher production of proinflammatory cytokines, that may limit the elimination of infected cells but drive immunopathology and higher disease severity during secondary infections [49,247,248]. Accordingly, a pre-existing proinflammatory pattern of T-cells is associated with the subsequent development of severe secondary DENV infection [249], whereas the expression of PD-1 by DENV-specific CD8^+^ T-cells is a correlate of protection associated with anti-viral mechanisms [86], suggesting that regulation of excessive immune responses is critical for disease resistance. Thus, a long-lasting protective role of T-cells against DENV is associated with an efficient T-cell priming by antigen-presenting cells expressing particular HLA molecules, modulation of bystander activation, and maintenance of high affinity antigen-specific IFN-γ-producing memory T-cells. However, the death of high-affinity DENV-specific T-cells induces the predominance of low-affinity, bystander-activated T-cells (or serotype cross-reactive T-cells in the case of DENV secondary infections), that produce proinflammatory cytokines and have low IFN-γ production, inducing endothelial dysfunction, vascular leakage, and organ damage (Figure 4). This model may apply for other VHF, where a massive bystander activation and apoptosis of antigen-specific T-cells is observed.

## 5. Conclusions

The interplay between virus and host is an important determinant in the pathogenesis and clinical outcome of VHF. T-cells are critical for the control of human VHF. The magnitude of their response is related to disease severity, survival, and protection. The induction of an efficient T-cell response is also a correlate of vaccine-induced protection. Thus, research efforts should focus on the study of mechanisms of protection or defective immune responses and fatal outcome, the immunodominant and preferentially cross-protective epitopes, and the relevant subsets in the human T-cell response against VHF. Activation markers such as HLA-DR, CD38, and Ki-67, are important predictors of the kinetics of the T-cell response during acute infection, and valuable for vaccine studies. The interaction between B-cells and T-cells for the generation of a protective antibody response is also important for the improvement of current vaccine candidates.

## Figures and Tables

**Figure 1 vaccines-07-00011-f001:**
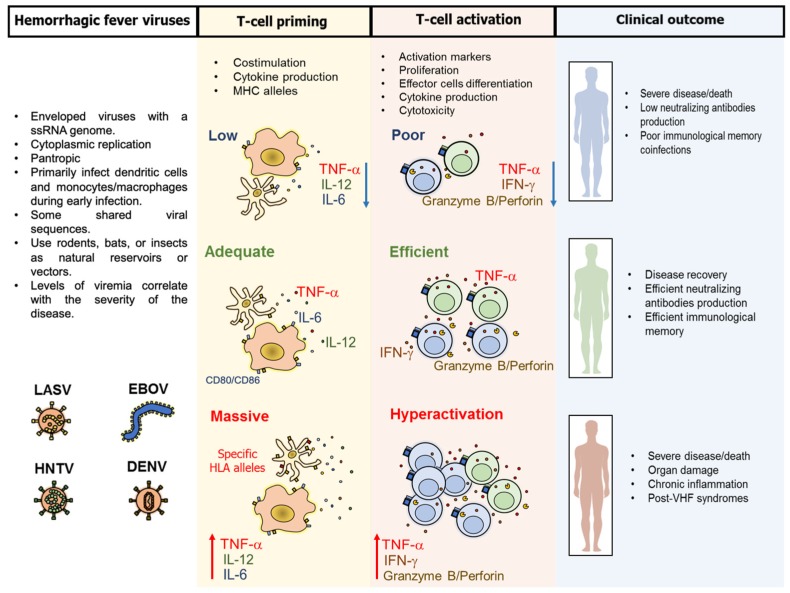
Common characteristics of viral hemorrhagic fever and the induced T-cell response. Lassa virus (LASV), ebola virus (EBOV), Hantaan virus (HNTV), and dengue virus (DENV), which share some virologic and epidemiological characteristics, primarily target dendritic cells and monocytes/macrophages during early infection, and induce three types of T-cell responses: 1. A low expression of costimulatory molecules (CD80/CD86), cytokine production and/or presentation of non-dominant epitopes, induces a poor T-cell activation, with low proliferation, low interferon (IFN)-γ and tumor necrosis factor (TNF)-α production and reduced cytotoxic potential. These defects can be responsible for severe disease/death, low induction of neutralizing antibodies, poor immunological memory and increased susceptibility to coinfections. 2. An efficient costimulation, cytokine production, and presentation of relevant epitopes under the context of protective HLA alleles, leads to optimal T-cell activation, which is reflected in disease recovery, induction of neutralizing antibodies and long-term immunological memory. 3. A massive costimulation, inflammatory cytokine production and expression of epitopes restricted by non-protective HLA alleles, leads to T-cell hyperactivation, with increased production of inflammatory cytokines that can lead to severe disease/death, organ damage, chronic inflammation and possibly to post-VHF syndromes.

**Figure 2 vaccines-07-00011-f002:**
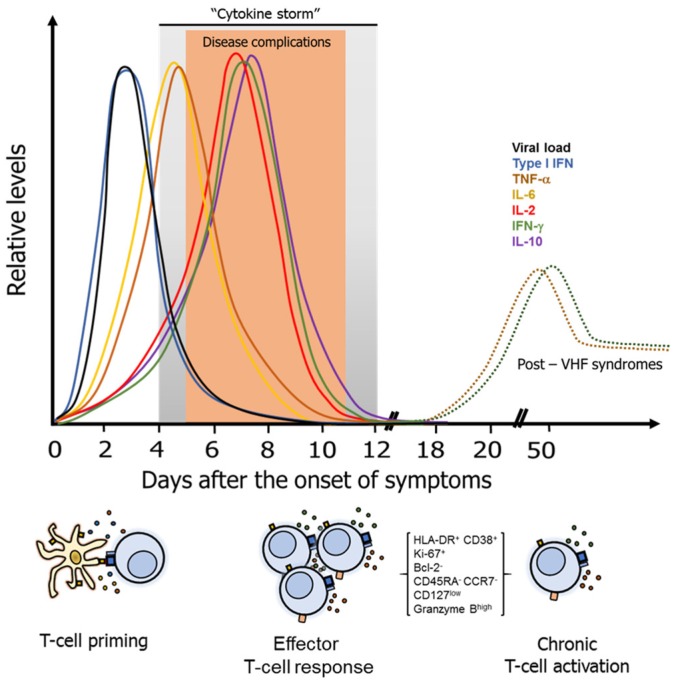
Kinetics of the cytokine response during VHF. Early during the acute phase of VHF, antigen-presenting cells produce high levels of type I IFN, and the inflammatory cytokines TNF-α and IL-6, that induce the activation of T-cells. This phase coincides with high levels of viremia that decreases after the fourth to fifth day after the onset of symptoms. The next period (from day 5 to 10) is characterized by the development of most disease complications, such as shock, hemorrhage and organ damage (shown as an orange region), coinciding with the massive activation of T-cells, and the increase in the levels of IFN-γ, IL-2, and IL-10. The period between days 4 to 12 is characterized by a massive cytokine production known as “cytokine storm” (shown as black line). Finally, the convalescence period coincides with the decrease in cytokine levels. In some VHF, the antigen persistence or perhaps epigenetic changes result in chronic T-cell activation, with increased levels of TNF-α and IFN-γ that are thought to be responsible for the development of post-VHF syndromes.

**Figure 3 vaccines-07-00011-f003:**
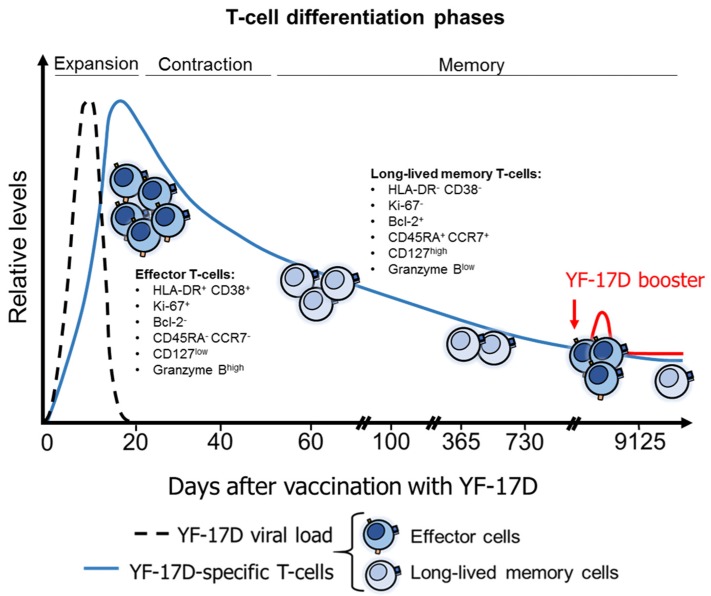
Dynamics of the human T-cell response after YF-17D vaccination. YF-17D replicates and exhibits a transient viremia, which stimulates the expansion of antigen-specific effector T-cells. This phase is followed by a contraction of the T-cell response, with massive death of effector populations. Finally, long-lived memory T-cells are observed after two to three months post-vaccination and remain detectable for at least 25 years. These memory populations are epigenetically poised to exert effector functions, and, after antigen re-stimulation, such as YF-17D boosters, a rapid differentiation of the effector profile is observed. However, the magnitude and quality of the remaining long-lived memory subsets are not influenced by vaccine boosters.

**Figure 4 vaccines-07-00011-f004:**
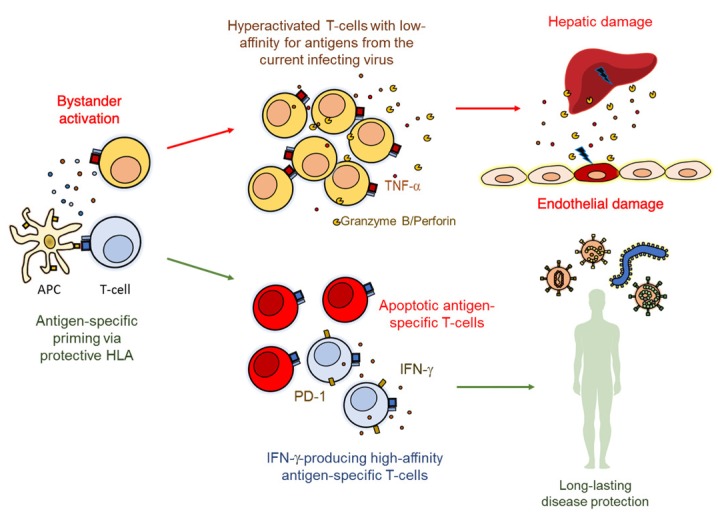
T-cell associated mechanisms of protection or susceptibility during VHF. T-cells can be activated in an antigen-specific or bystander manner, the latter through inflammatory cytokine signals. Bystander activation results in an exacerbated response of low-affinity cross-reactive T-cells (particularly during secondary infections of antigenically related viruses), that produce high levels of tumor necrosis factor (TNF)-α and other inflammatory cytokines and cytotoxic molecules, that associate with endothelial and/or hepatic damage. Antigen-specific priming of T-cells under the context of protective HLA alleles can lead to the generation of highly functional antigen-specific T-cells, which are characterized by the expression of inhibitory receptors such as programmed death (PD)-1, that can modulate their response and prevent hyperactivation. Individuals who maintain high frequencies of antigen-specific T-cells, despite increased apoptosis of these subsets, can preserve long-lasting immunity and protection against subsequent related viral infections.

**Table 1 vaccines-07-00011-t001:** Common T-cell features among VHF entities.

Feature	Viral Hemorrhagic Fever Families				References
	*Arenaviridae* LASV	*Filoviridae* EBOV	*Hantaviridae* Hantaviruses	*Flaviviridae* DENV	
**Most prominent T-cell role**	More protective than pathogenic	More protective than pathogenic	More protective than pathogenic	More pathogenic than protective	[33,43,45,46]
T-cell immunodominant viral protein	Nucleoprotean glycoprotein	Nucleoprotein	Nucleocapsid protein; glycoprotein	Non-structural proteins	[45,47,48,49]
Reported effector protective mechanisms	IFN-γ, TNF-α, IL-2	IFN-γ, TNF-α, production	IFN-γ, TNF-α, production cytotoxicity	TNF-γ, production cytotoxicity	[33,46,50,51]
Apoptosis of T-cells	Yes	High	Yes	Yes, particularly of antigen-specific cells	[52,53,54,55]
Defective T-cell activation or dysfunction	Low activation capacity by antigen-presenting cells; low proliferative and cytokine T-cell response	Low activation capacity by antigen-presenting cells	Not reported Hantavirus nucleocapsid protein inhibits granzyme B-mediated apoptosis	Low activation capacity by infected antigen-presenting cells	[56,57,58]

**Table 2 vaccines-07-00011-t002:** Activation and differentiation markers used for the study of T-cell responses after natural and vaccine challenges.

Molecule	Function	T-Cell Subset Marker
HLA-DR	Class II major histocompatibility complex molecule	Effector cells
CD38	ADP-ribosyl cyclase ectoenzyme	Effector cells
Ki-67	Nuclear protein expressed during active phases of cell cycle and associated with cell proliferative activity	Effector cells
B-cell lymphoma (Bcl)-2	Anti-apoptotic protein that prevents the release of cytochrome C and oxygen reactive species from mitochondria	Naïve and central memory cells
CCR7	Chemokine receptor for the CCL19 and CCL21 chemokines. Involved in the homing of T-cells to secondary lymphoid organs	Naïve and central memory cells
CD45RA	Phosphatase involved in T-cell receptor signal transduction	Naïve and central memory cells and terminally-differentiated cells
CD45RO	Phosphatase involved in T-cell receptor signal transduction	Memory cells
CD28	Receptor for costimulatory molecules, involved in the amplificationof T-cell receptor signalir	Naïve and memory cells, decreasing cell differentiation
CD127	Alpha chain of the II-7-receptor; IL-7 supports the survival of mature T-cells	Naïve and memory cells, decreasing along cell differentiation
Programmed death 1 protein (PD-1)	Regulatory receptor that inhibits T-cell receptor signaling	Memory cells
Perforin/Granzyme B	Cytotoxic molecules induce apoptosis intarget cells	Effector cells

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
