# Peer review of "T-Cell Response to Viral Hemorrhagic Fevers"

_vaccines, 2019, doi:10.3390/vaccines7010011_

Round 1

Reviewer 1 Report

This review by Perdomo-Celis et al. is well-written and addresses an important and often-overlooked topic.  The manuscript is well-referenced and attempts to draw an overview of T cell responses across multiple viral hemorrhagic fevers.  While this is useful, modifications to the manuscript would help it be more nuanced in its conclusions.

Major concerns:

There are many unknown aspects to T cell responses in viral hemorrhagic fevers, even within viral families.  Therefore, it is too simplistic to broadly characterize T cell responses as done in Table 2.  For example, it is simply unknown whether filovirus-infected individuals have "Massive T cell activation" as listed in Table 2; the literature is mixed, but overall suggests that survivors have an early but controlled T cell response, whilst non-survivors have either a depressed T cell response OR a massively a massive T cell response that is delayed (see Speranza et al., JID 2018 PMID 29986035 and Bradfute et al., JI 2008 PMID 18322215 for examples). 

Similarly, the reference used in Table 2 to show that the dominant T-cell differentiation stage during acute infection for Ebola virus is "Effector and terminal effector memory" is a study that follows a single survivor long-term.  It is therefore difficult to tell if this is a) normal and b) different between survivors and fatalities (who may not have time to generate any memory responses).

The authors need to better distinguish between T cell responses in lethal and nonlethal infection, as well as between acute T cell responses and chronic responses.  There should also be a clearer discussion of responses in infection as compared to vaccination.  For example, lines 288-296 are confusing, since it is not clear whether these responses are found in survivors, nonsurvivors, vaccinees, etc.  This occurs throughout the manuscript.

Minor concerns

  Lines 282-283, specify that this is only in wild-type, not mouse-adapted, EBOV infection.

  Line 335, should "HPS" be changed to "HFRS?"

Author Response

Major concerns:

There are many unknown aspects to T cell responses in viral hemorrhagic fevers, even within viral families. Therefore, it is too simplistic to broadly characterize T cell responses as done in Table 2.  For example, it is simply unknown whether filovirus-infected individuals have "Massive T cell activation" as listed in Table 2; the literature is mixed, but overall suggests that survivors have an early but controlled T cell response, whilst non-survivors have either a depressed T cell response OR a massively a massive T cell response that is delayed (see Speranza et al., JID 2018 PMID 29986035 and Bradfute et al., JI 2008 PMID 18322215 for examples).  

Similarly, the reference used in Table 2 to show that the dominant T-cell differentiation stage during acute infection for Ebola virus is "Effector and terminal effector memory" is a study that follows a single survivor long-term.  It is therefore difficult to tell if this is a) normal and b) different between survivors and fatalities (who may not have time to generate any memory responses).

The authors need to better distinguish between T cell responses in lethal and nonlethal infection, as well as between acute T cell responses and chronic responses.  There should also be a clearer discussion of responses in infection as compared to vaccination.  For example, lines 288-296 are confusing, since it is not clear whether these responses are found in survivors, nonsurvivors, vaccinees, etc.  This occurs throughout the manuscript.

Response: In order to clarify the differences between patients with severe disease/death and those with clinical recovery, in Figure 1 we illustrate the characteristics of the T-cell response that could explain the different clinical outcomes. Similarly, in Figure 2 we present the cytokine and T-cell response during acute and chronic infections. For further understanding, as suggested by the reviewer, we have made the following changes in the manuscript to distinguish the T-cell response in fatal and non-fatal VHF and during acute and chronic infection:

The sections “Massive T-cell activation” and “Dominant T-cell differentiation stage during acute infection” in Table 1 were deleted to avoid generalization between different clinical scenarios.

The differences in the T-cell response between survivors and non-survivors was denoted for each viral entity. For example, see page 8, line 277 and 285; page 11, Lines 352-353.

 A discussion on the findings of the studies evaluating the T-cell features and the relation with the time of infection and disease survival was added on page 4, Line 132-139.

T-cell responses in the specific scenarios were specified throughout the text. For example, see page 10, Lines 341-344; page 15, lines 578 and 585-586.

Minor concerns

Lines 282-283, specify that this is only in wild-type, not mouse-adapted, EBOV infection.

Response: As suggested by the reviewer, it was specified that these viral challenges were performed with a mouse-adapted EBOV (page 10, line 342-344).

Line 335, should "HPS" be changed to "HFRS?"

Response: In agreement with the reviewer, “HPS” was changed for “HFRS” on page 11, line 399.

Reviewer 2 Report

Perdomo-Celis et.al., have provided a generalized overview of what is known about T-cell contributions to hemorrhagic fever biology and pathogenesis and how this information relates to some vaccine approaches.  The work is well written and appears to attempt to unify the role of this cell type to VHF disease and survival processes.  There are some point for clarification that would benefit the reader if suggested points were addressed.

  How do the T-cell responses addressed by these viruses differ from other ssRNA viruses?  Is it a generalized response to virus infection or is there something special about what happens with VHF viruses that would invite the need for a review like this.

I  think the authors should address the differences across the bunyaviridae family a little better by including CCHF and Rift as well.  This is particularly important/interesting due to the differences in vectors across viral types.  Could these differences also translate to differential T-cell responses?

An extension of the above comment, what of new world versus old world hantaviruses and arenaviruses?  The pathologies are not identical in either case, could there also be differential T-cell responses?  The authors touch on this, but unclear if this is the limit of what is contained in the literature or if perhaps more work should be done to "fill out" this concept.

Figure 1:  correct description of VHF viruses to denote that surveilling antigen presenting cells including dendritic and monocyte/macrophage cells are first cells to be infected, as written it is too suggestive that these are the only cells infected...which while mostly true early in infection, is most certainly not true later as endothelium and hepatocytes are heavily impacted as the disease progresses, some more than others.

Figure 1:  Please include key for costimulatory molecules/cytokines or more clearly demarcate.

Lines77-78:  1-5% mortality does not adjust appropriately for cases that present to hospital which can dramatically increase the CFR and further differences may be found between Mano-River Union Area and Nigeria.  https://journals.plos.org/plosntds/article?id=10.1371/journal.pntd.0002748

Line81:  Update this to account for present case load and CFR or include text "at the time of this writing"

Line83:  Probably better to say North-Eastern Europe as there are a number of cases in Scandanavian and German localities as well.

Line 84:  Include Canada as site for HPS cases.

Line94: Current estimates on number of cases, this report is from 6 years ago...is it the same?

Line 126:  Change text to:  Particularly, T-cells have been purported to contribute to....or similiar...

Line 139-147:  Plenty of evidence supporting cytokine production by other cell types (endothelial cells, neutrophils, NK, hepatocytes, ect....), please adjust or mute this section to avoid over-stepping claim here.  Also, its not only cytokines that lead to systemic inflammatory response (aka cytokine storm), i.e. nitrogen and oxide radicals, proinflammatory lipids, ect...

Line 598...better to say "associated with endothelial and/or hepatic damage"  there are too many players involved to nail it down to T-cells gone haywire alone...

Figure 4:  Provide a symbol key for the different things floating around in "hyperactiveated t-cell" part of figure.

Author Response

How do the T-cell responses addressed by these viruses differ from other ssRNA viruses?  Is it a generalized response to virus infection or is there something special about what happens with VHF viruses that would invite the need for a review like this.

Response: As suggested by the reviewer, we have included a discussion about differences between hemorrhagic fever viruses and other clinically relevant ssRNA viruses, on page 6, lines 179-195.

I think the authors should address the differences across the bunyaviridae family a little better by including CCHF and Rift as well.  This is particularly important/interesting due to the differences in vectors across viral types.  Could these differences also translate to differential T-cell responses?

Response: As suggested by the reviewer, we have included a discussion about the differences between bunyavirales CCHF and RFV. Of note, the available data indicate that a similar T-cell response is present in these infections, compared with hantaviruses. See page 11, lines 410-418.

An extension of the above comment, what of new world versus old world hantaviruses and arenaviruses?  The pathologies are not identical in either case, could there also be differential T-cell responses?  The authors touch on this, but unclear if this is the limit of what is contained in the literature or if perhaps more work should be done to "fill out" this concept.

Response: As suggested by the reviewer, we included a discussion on the differences between Old and New World arenaviruses (page 8, line 280-285). In addition, the available data do not reflect differences between the T-cell response between New World and Old World hantaviruses. A sentence regarding this issue was added in page 11, line 416-418.

Figure 1:  correct description of VHF viruses to denote that surveilling antigen presenting cells including dendritic and monocyte/macrophage cells are first cells to be infected, as written it is too suggestive that these are the only cells infected...which while mostly true early in infection, is most certainly not true later as endothelium and hepatocytes are heavily impacted as the disease progresses, some more than others.

Response: As suggested by the reviewer, the description in Figure 1 was edited as follows: “Primarily infect dendritic cells and monocytes/macrophages during early infection” (page 2, line 48-49).

Figure 1:  Please include key for costimulatory molecules/cytokines or more clearly demarcate.

Response: As suggested by the reviewer, a key for costimulatory molecules was added in Figure 1 and other keys were more clearly demarcated.

Lines77-78:  1-5% mortality does not adjust appropriately for cases that present to hospital which can dramatically increase the CFR and further differences may be found between Mano-River Union Area and Nigeria.  https://journals.plos.org/plosntds/article?id=10.1371/journal.pntd.0002748

Response: In agreement with the reviewer, we have included a distinction of the case fatality rate among patients assisted in health care facilities (page 3, lines 79-80).

Line81:  Update this to account for present case load and CFR or include text "at the time of this writing"

Response: As suggested by the reviewer, case load was updated (page 3, line 82-82).

Line83:  Probably better to say North-Eastern Europe as there are a number of cases in Scandanavian and German localities as well.

Response: As suggested by the reviewer, “Russia” was changed to “North-Eastern Europe” (page 3, line 85).

Line 84:  Include Canada as site for HPS cases.

Response: As suggested by the reviewer, Canada was included as the site for HPS cases (page 3, line 86).

Line94: Current estimates on number of cases, this report is from 6 years ago...is it the same?

Response: As suggested by the reviewer, an update in the cases of YF in Africa was added in page 3, lines 97-99.

Line 126:  Change text to:  Particularly, T-cells have been purported to contribute to or similiar...

Response: As suggested by the reviewer, page 4, line 130 has been edited.

Line 139-147:  Plenty of evidence supporting cytokine production by other cell types (endothelial cells, neutrophils, NK, hepatocytes, ect....), please adjust or mute this section to avoid over-stepping claim here.  Also, its not only cytokines that lead to systemic inflammatory response (aka cytokine storm), i.e. nitrogen and oxide radicals, proinflammatory lipids, ect...

Response: As suggested by the reviewer, other cytokine-producing cells and soluble factors contributing to the cytokine storm were included in page 6, line 201-203.

Line 598...better to say "associated with endothelial and/or hepatic damage" there are too many players involved to nail it down to T-cells gone haywire alone...

Response: As suggested by the reviewer, page 17, line 673 has been edited.

Figure 4:  Provide a symbol key for the different things floating around in "hyperactiveated t-cell" part of figure.

Response: As suggested by the reviewer, a symbol key was added In Figure 4.

Round 2

Reviewer 1 Report

Modifications are sufficient to address this reviewer's concerns.